# An Integrated Approach for Constraining Depositional Zones in a Tide-Influenced River: Insights from the Gorai River, Southwest Bangladesh

**Edwin J. Bomer [1,2,\*], Carol A. Wilson [1,2] and Dilip K. Datta [3]**

[1] Department of Geology and Geophysics, Louisiana State University, Baton Rouge, LA 70803, USA; carolw@lsu.edu
[2] Coastal Studies Institute, Louisiana State University, Baton Rouge, LA 70803, USA
[3] Environmental Science Discipline, Khulna University, Khulna 9208, Bangladesh; dkd_195709@yahoo.com
[\*] Correspondence: ebomer1@lsu.edu; Tel.: +1-361-876-2625

**Abstract:** The tidal to fluvial transition (TFT) of estuaries and coastal rivers is one of the most complex environments on Earth with respect to the transportation and deposition of sediment, owing in large part to competing fluvial and marine processes. While there have been recent advances in the stratigraphic understanding of the TFT, it is still unclear whether these findings are site-specific or representative of mixed tidal-fluvial systems worldwide. Yet, research from this depositional domain holds profound societal and economic importance. For instance, understanding the underlying stratigraphic architecture of channel margins is critical for assessing geomorphic change for fluvio-deltaic settings, which are generally vulnerable to lateral channel migration and resultant erosion. Findings would also benefit paleo-geographic reconstructions of ancient tide-influenced successions and provide an analog for hydrocarbon reservoir models. In the Ganges-Brahmaputra Delta of Bangladesh, the Gorai River is one of two Ganges distributaries actively connected to the Bay of Bengal. With fluvial input from the Ganges and meso-scale (2–4 m range) tides at the coast, the Gorai exhibits a variety of hydrodynamic regimes across its 350-km reach, providing a unique opportunity to investigate along-channel depositional patterns across the TFT. This study integrates multiple datasets—core sedimentology, river channel bathymetry, and remote sensing—to provide a process-based framework for determining the relative position of sedimentary deposits within the tidal-fluvial continuum of the Gorai River. The results of this investigation reveal coincident, abrupt shifts in river channel morphology and sediment character, suggesting the occurrence of backwater-induced mass extraction of relatively coarse sediments (i.e., fine sand). Despite being situated in an energetic tidal environment, evidence of tidal cyclicity in cored sediments is relatively rare, and the bulk stratigraphy appears strongly overprinted by irregularly spaced cm- to dm-scale sediment packages, likely derived from monsoonal flood pulses. Such findings differ from previously-studied mixed tidal-fluvial systems and underscore the site-specific complexities associated with this depositional domain.

**Keywords:** tidal-fluvial transition; hydrodynamics; sedimentology; seasonal; Bangladesh

## 1. Introduction

Rivers, estuaries, and other transitional bodies of water are subjected to varying degrees of terrestrial and marine influences on their journey from headwater source to open ocean, resulting in a complex spectrum of hydrodynamic and depositional conditions (e.g., Ref. [1]). In landward regions, river currents, particularly those produced during flood events, represent the dominant hydrodynamic control on sediment transport and deposition (e.g., Refs. [2,3]). However, as rivers approach the coast,

their currents are tempered by waning hydraulic gradients and the increasing influence of marine processes, namely waves and tides. The impact of waves is typically limited to the coastal zone as wave energy rapidly attenuates with distance inland through bottom friction [4,5] and interaction with vegetation [6–8]. Tides, on the other hand, can propagate upstream for tens to hundreds of kilometers, depending on river discharge, channel dimensions, and the gradient of the channel bed (e.g., Ref. [9]). Rivers connected to active tidal coasts are therefore characterized by reaches of tidal dominance, mixed influence, and fluvial dominance, together referred to in the literature as the tidal to fluvial transition zone (TFT, e.g., Ref. [10]).

How this continuum of hydrodynamic processes controls the spatial distribution and preservation of sediments along tide-influenced rivers has only recently been investigated and is largely focused on interpretations from ancient tidal strata (e.g., Refs. [11–17]). Comparatively little work has been undertaken in the TFT of modern systems [18–21], in spite of its dual applicability to economic and environmental issues. For instance, core data from modern tide-influenced channel bars readily allows the connection of hydraulic process to sedimentary product, which has proven useful for characterizing tidal-fluvial hydrocarbon reservoirs such as the Cretaceous McMurray Formation of Canada [22] and the Triassic Mungaroo Formation of Australia [17]. Constraining the subsurface abundance of cohesive (i.e., mud) and non-cohesive (i.e., sand) sediments in the TFT is also critical for predicting geomorphic change along low-lying coastal rivers. Such findings, in turn, can guide the management of environmental and societal matters like groundwater resources [23], aquaculture [24], channel infilling and land reclamation [25], and adaptation to lateral river migration and erosion [26], all of which are pertinent to communities residing in fluvio-deltaic landscapes.

Previous research in both modern and ancient tidal-fluvial settings has demonstrated the importance of utilizing multiple proxies to accurately interpret and characterize depositional conditions (e.g., Refs. [15,20,27]). The foundation of these studies has been sedimentological trends, namely changes in bulk grain size and stratal cyclicity as determined through sediment coring, outcrop observations, and channel bed grab sampling. Trace fossil assemblages (ichnofacies) have often been used to corroborate sedimentological findings, whereby faunal burrow size, intensity, and diversity relate to the level of physiochemical stress induced by the environment, therefore providing a proxy for salinity and marine influence [28–30]. Separate efforts have been made to understand how the transition from backwater to normal flow hydrodynamics governs sediment transport [31,32] and river channel kinematics [33–36]. However, only preliminary relationships between backwater hydrodynamics and channel margin stratigraphy exist [35], and these have not been analyzed within the context of the TFT.

Among studies of tide-influenced rivers worldwide, much attention has been centered on systems in North America, notably the Fraser River (e.g., Refs. [19,27,29,37,38]). Knowledge gaps exist elsewhere, especially in South Asia, a region that simultaneously hosts the largest tide-dominated deltas (e.g., Ganges-Brahmaputra, Indus, Ayeyawady) and population centers on Earth [39,40]. The present study seeks to fill this gap by considering the mixed tidal-fluvial Gorai River, a distributary of the Ganges River in the Ganges-Brahmaputra Delta of Bangladesh (Figure 1). The main objective of this study is to integrate a diverse assemblage of datasets, including core sedimentology, remote sensing, and channel bathymetry to provide a process-based depositional framework for the TFT of the Gorai River. The Gorai differs from many tidal-fluvial rivers in that its hydrology is highly regulated by seasonal conditions, exhibiting an order of magnitude difference in water discharge between wet and dry seasons [41]. Despite the growing body of literature in the TFT, very few studies have investigated systems with such pronounced hydrologic variation [20], and it is presently unclear if and how these seasonal signals are preserved in the stratigraphy. Another key objective of the study is to explore the interactions among channel gradient, hydraulic conditions, and channel bank composition in modulating river morphodynamics. Field and experimental work have demonstrated that the upstream limit of the backwater transition, that is the location where the channel bed elevation approximates mean sea level [42], exerts a strong control on the positioning of river avulsions (e.g., Refs. [31,35,43–46]), and it is hypothesized that similar phenomena occur in the Gorai River. The identification of areas

likely to undergo morphologic change would be particularly relevant for southwest Bangladesh but would also contribute observational data to global modelling efforts that forecast river avulsion location and frequency (e.g., Refs. [33,47]).

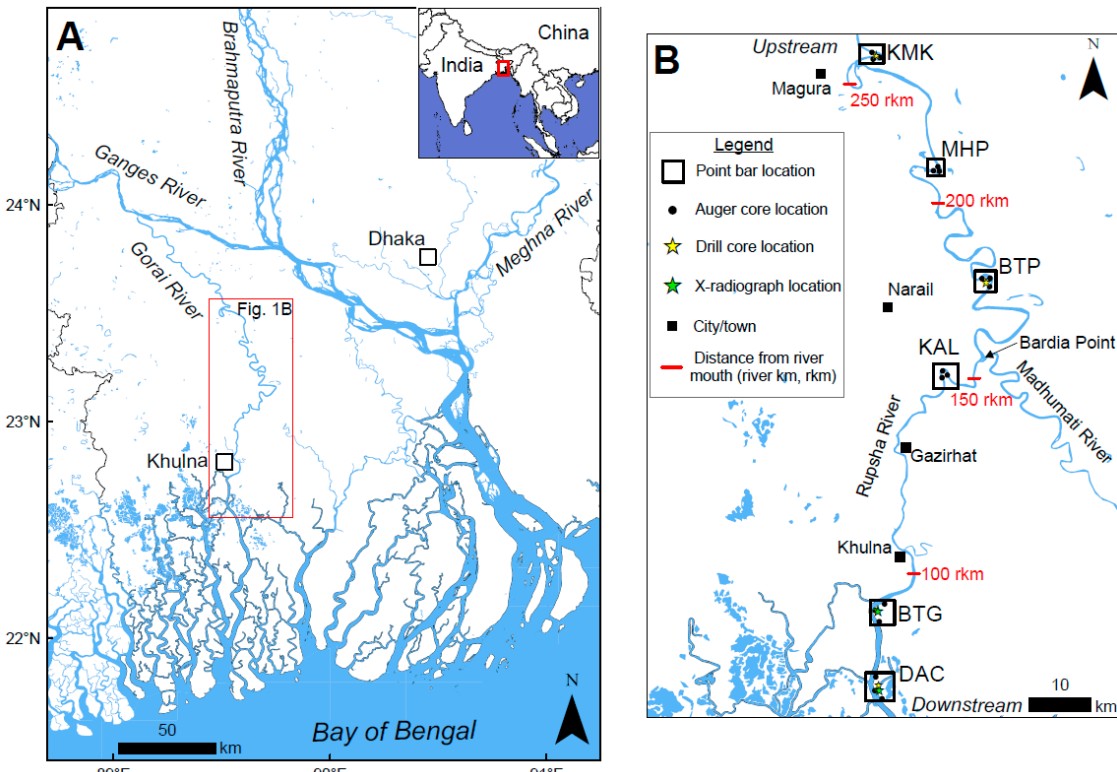

**Figure 1.** (**A**) Geographic context of the study area within greater Bangladesh and Southeast Asia. The Gorai River is influenced by fluvial forces (e.g., monsoonal flood discharge) from the Ganges and marine forces (e.g., meso-scale tides) from the Bay of Bengal. (**B**) Map of the main stem of the Gorai River illustrating sampling locations codified by the names of nearby villages: Dacope (DAC), Batiaghata (BTG), Kalia (KAL), Bhatiapara (BTP), Mohammadpur (MHP), and Kamarkhali (KMK). Note that between Bardia Point and Khulna, the Gorai is locally referred to as the Rupsha River. Downstream of Khulna the Gorai is locally known as the Pussur River.

## 2. Regional Setting

The Gorai River is one of the largest distributaries of the Ganges and comprises the principal source of fresh river water for southwest Bangladesh, a region that hosts a burgeoning population of approximately 30 million [48]. Originating from the Ganges in Kushtia District, the Gorai flows south-southeast for ~350 km through Quaternary alluvial and tidal delta plain sediments before emptying into the Bay of Bengal (Figure 1A). In the upper reaches, the river exists as a single, meandering trunk channel with no natural bifurcations or distributaries (Figure 1B). At Bardia Point, located ~150 river km upstream of the mouth, the Gorai bifurcates into the Rupsha and Madhumati Rivers (Figure 1B). Historically, the Madhumati accommodated the majority of discharge from the Gorai until the beginning of the 20th century [49], when the Rupsha off-take was extensively dredged to augment the delivery of river water to Khulna [50]. Since then, the Rupsha has taken precedence as the primary distributary, while the Madhumati has degraded as a result of waning fluvial input and channel siltation: Between 1965 and 1998, the average depth of the Madhumati River decreased 55% from 4.86 to 2.76 m [50]. Downstream of Khulna, the Gorai-Rupsha enters the tidal delta plain and is locally referred to as the Pussur River. Here, the Pussur branches into a dense and interconnected network of tidal channels surrounding mangrove-vegetated forest (Sundarbans National Forest). For simplicity, the Gorai, Rupsha, and Pussur reaches of the river are hereafter referred to collectively as

the Gorai River. In total, the drainage basin of the Gorai and its distributaries encompass ~16000 km$^2$, roughly half of which is cultivated land [50].

Similar to other tropical rivers in South and Southeast Asia, water discharge of the Gorai exhibits dramatic seasonality and is largely controlled by monsoonal precipitation and flooding. For instance, at the Kamarkhali gauging station located ~100 river-km downstream of the Ganges off-take ("KMK" in Figure 1B), ~95% of the annual discharge occurs between the months of July and November [41]. Due to the extreme variability of water discharge, the entire course of the river is fresh during the monsoon season but is saline in the dry season as far upstream as Gazirhat (see Figure 1B for location [51]). The upstream extent of tidal influence is similarly impacted by the magnitude of discharge: In the dry season, tides (range ~ 0.2 m) are observed up to Mohammadpur ("MHP" in Figure 1B), whereas during the monsoon season, tidal influence is dampened by the flood pulse and is limited to ~20 river km downstream of MHP (Figure 1B; [50]). The lower reach of the Gorai is characterized by a hypersynchronous tidal regime, wherein the tide range increases inland from ~2 m at the coast until reaching a maximum of ~3 m at Dacope ("DAC" in Figure 1B; [50]). This study is focused on a ~200-km reach of the Gorai River extending from a non-tidal, perennially fresh-water setting at KMK to a tide-dominated, seasonally-saline setting at DAC (Figure 1B).

## 3. Methods

### 3.1. Remote Sensing

A digital elevation model (DEM) of the study area was constructed using data from the 2000 Shuttle Radar Topography Mission (SRTM; [52]). SRTM elevation data have a spatial resolution of 1 arc-second (~30 m) and were downloaded from the open source USGS EarthExplorer database (https://earthexplorer.usgs.gov/). Post-processing of the DEM and creation of elevation transects adjacent to the Gorai River were carried out in the software ArcMap 10.5. Regional trends in elevation change were identified using a running average (10 km window) of the raw data.

Present-day river channel sinuosity was calculated for each meander bend along the Gorai River by dividing the channel length by the linear downstream length (e.g., Ref. [53]). The lateral mobility of point bars was determined by applying the "best-fitting circle" method [54,55] to georeferenced Landsat aerial photographs from 1972 and 2015 (resolution of 90 and 30 m, respectively), also obtained from the EarthExplorer database. This approach accounts for changes in meander bend extension ($\delta E$) by comparing bend radii ($R_b$) at two points in time (t and t + $\delta$t), represented by Equation (1).

$$\partial E = R_b(t + \partial t) - R_b(t) \tag{1}$$

Bar migration via downstream translation ($\delta T$) is also calculated by comparing the change in location of the bend center point coordinates ($\delta x$ and $\delta y$), represented by Equation (2).

$$\partial T = \sqrt{(\partial x_c)^2 + (\partial y_c)^2} \tag{2}$$

The overall point bar migration rate ($M_b$), which combines the extension ($\delta E$) and translation ($\delta T$) components, is then calculated using the following equation

$$M_b = \sqrt{\partial R^2 + \partial S^2 / (\partial t)} \tag{3}$$

### 3.2. Hydrodynamic Observations

Channel bathymetry of the Gorai River [56] was incorporated with elevation measurements to constrain the vertical position of the channel bed with respect to mean sea level. The bathymetric survey consists of 183 bank-normal transects with ~1 km spacing between each transect. However, this study only utilizes 50 of the most inland transects, covering a segment of the river from Bardia Point to

Dacope (DAC, Figure 1B). Bathymetric data processing and presentation followed an approach similar to that of Nittrouer et al. [31]. Briefly, the average channel depth of each transect was calculated, and then a 10 km running average was applied to the trend to neutralize the effects of localized bathymetric irregularities, such as bed scouring from channel confluences (e.g., Ref. [57]). Bathymetric data were not corrected for water level changes induced by tides or river discharge.

Time-averaged hydrodynamic regions of the Gorai River were determined by analyzing annual-scale water level data from tide gauging stations across the TFT [50,58] and salinity intrusion maps [51]. Following a convention similar to that of La Croix and Dashtgard [27], tide-dominated or "tidal" reaches of the river were classified as having a mean tide range ≥ 2 m (i.e., meso-tidal or greater) and seasonal polyhaline conditions (18 to 30 psu); mixed tidal-fluvial or "mixed" reaches were classified as having a mean tide range ≥ 0.2 m but ≤ 2 m (i.e., micro-tidal) and seasonal mesohaline conditions (5 to 18 psu); and fluvial-dominated, or "fluvial," reaches were classified as having a mean tide range of ≤ 0.2 m (i.e., non-tidal) and perennial fresh water conditions (<0.5 psu).

### 3.3. Site Selection and Fieldwork

Core collection was undertaken during four field campaigns: following the summer monsoon in October 2017 and 2018, and during the dry season in March 2018 and 2019. Six point bars across the Gorai River TFT were cored in transects of ~1 km with ~300 m spacing between core sites (Figure 1B). Sampling locations were pre-selected using satellite imagery and verified in the field with a Garmin eTrex 10 global positioning system (GPS) referenced to the WGS84 datum. Cores up to ~3 m length were extracted using a 3-cm diameter half-cylinder auger driven into the subsurface using an iron sledge. Sediment cores were taken along the accreting perimeter of the point bar with sites situated on the upstream (U), apex (A), and downstream (D) sides to capture along-strike depositional changes. In addition, acrylic box cores (30 cm length) were collected from DAC and BTG locations for x-radiograph imaging. To ensure consistent lateral positioning relative to the water line, all cores were collected following the monsoon season (i.e., during high stage) in upstream locations and during or near high tide in tide-influenced areas. Following extraction, all auger cores were transported to the Environmental Science Discipline at Khulna University where they were described and subsampled in regular intervals.

To constrain deeper stratigraphy and its effect on channel mobility, drill cores to ~45 m depth were taken at DAC, BTP and KMK locations (Figure 1B), representing tidal, mixed tidal-fluvial, and fluvial environments, respectively. Drilling was carried out using a local method for installing tube wells, previously described by Pickering et al. [59]. In brief, a drill string fitted with a steel cutting shoe was driven into the subsurface using a bamboo fulcrum-and-lever system facilitated by circulating drill fluid (water and organic matter). Sediment cuttings were collected from the expelled drill fluid in 1-m depth intervals for further analyses.

### 3.4. X-Radiography

X-radiographs of sediments encased in acrylic box cores were collected to reveal sedimentological (e.g., bedding and laminations) and biological (e.g., burrows) features that were not visible otherwise. X-radiographs were carried out using a medical-grade X-ray detector (Khulna Health Diagnostic Center, Khulna, Bangladesh) illuminated by an X-ray unit operating at 40 keV and 10 mA. Individual x-radiograph films were developed and then photographed over a light table to maximize visibility. The intensity of bioturbation as revealed by x-radiographs was reported following the classification scheme of Taylor and Goldring [60], wherein bioturbation index (BI) is quantified on a scale from 0 to 6. Lower values (0 to 3) represent high preservation of original sedimentary fabric and little to no bioturbation, while higher values (4 to 6) indicate extensive biological overprinting and sediment reworking [60].

*3.5. Granulometry*

Grain size analysis was conducted at 10-cm intervals for all auger cores and at a finer resolution when distinct changes in sediment character were noted (n = 538). For deeper drill cores, analyses were undertaken at 1-m intervals (n = 135). In each case, sediment samples of ~2 g were exposed to 2 mL of 30% hydrogen peroxide ($H_2O_2$) to eliminate fine organic matter. Digested solutions were then stirred with 15 mL of 0.05% sodium metaphosphate ($NaH_2PO_4$) to de-flocculate clay particles. Prior to analyses, particulate organic debris and clastic material larger than 850 µm were removed by sieving. A total of 678 samples were analyzed for volumetric frequency distributions of grain size (range = 0.4-850 µm) and particle sorting using a Beckman Coulter laser diffraction particle size analyzer (Model LS 13 320).

## 4. Results

*4.1. Surface and Channel Morphology*

The surface elevation of the Gorai River floodplain gradually increases landward, ranging from 3 m immediately north of the Sundarbans Forest to 15 m near the Ganges-Gorai offtake (Figures 2 and 3). However, closer examination reveals two reaches with distinctly different gradients (sensu [61]; Figure 3). The first reach, which extends from the beginning of the transect ("A" in Figures 2 and 3) to ~10 km upstream of KAL, exhibits an average slope of $5 \times 10^{-6}$ (Figure 3). Inland of this point, the average slope of the floodplain increases by over an order of magnitude to $1 \times 10^{-4}$ (Figure 3). Integration of channel bathymetry [56] with the elevation data indicates that the channel bed is situated below mean sea level (i.e., in the backwater zone) until approximately ~130 river km inland of the coast (Figures 1B, 2 and 3).

Although present-day channel sinuosity and channel migration rates (measured between 1972 and 2015) both generally increase landward, more detailed trends emerge when these geomorphic parameters are placed within the context of the hydrodynamic zones of the river (Figure 4). In the downstream, tide-dominated tract of the river, channel sinuosity is relatively low and exhibits little variation along its course, with an average sinuosity coefficient (±standard deviation) of 1.2 ± 0.1 (Figure 4A). Migration rates in the tide-dominated realm are similarly low, exhibiting an average migration rate of 6.6 ± 2.0 m/yr (Figure 4B). Channel sinuosity and migration rates dramatically increase across the backwater (i.e., into the mixed tidal-fluvial zone), demonstrating a 50% increase in the sinuosity coefficient to 1.8 ± 0.6 and a nearly five-fold increase in migration rates to 32.4 ± 16.1 m/yr (Figure 4A,B). In the purely fluvial zone, beyond the upstream limit of tidal influence, channel sinuosity is roughly the same as in the mixed tidal-fluvial zone (1.7 ± 0.9, Figure 4A). However, channel migration rates decrease to 9.0 ± 11.4 m/yr, approximately the same level as the tide-dominated reach of the river (Figure 4B).

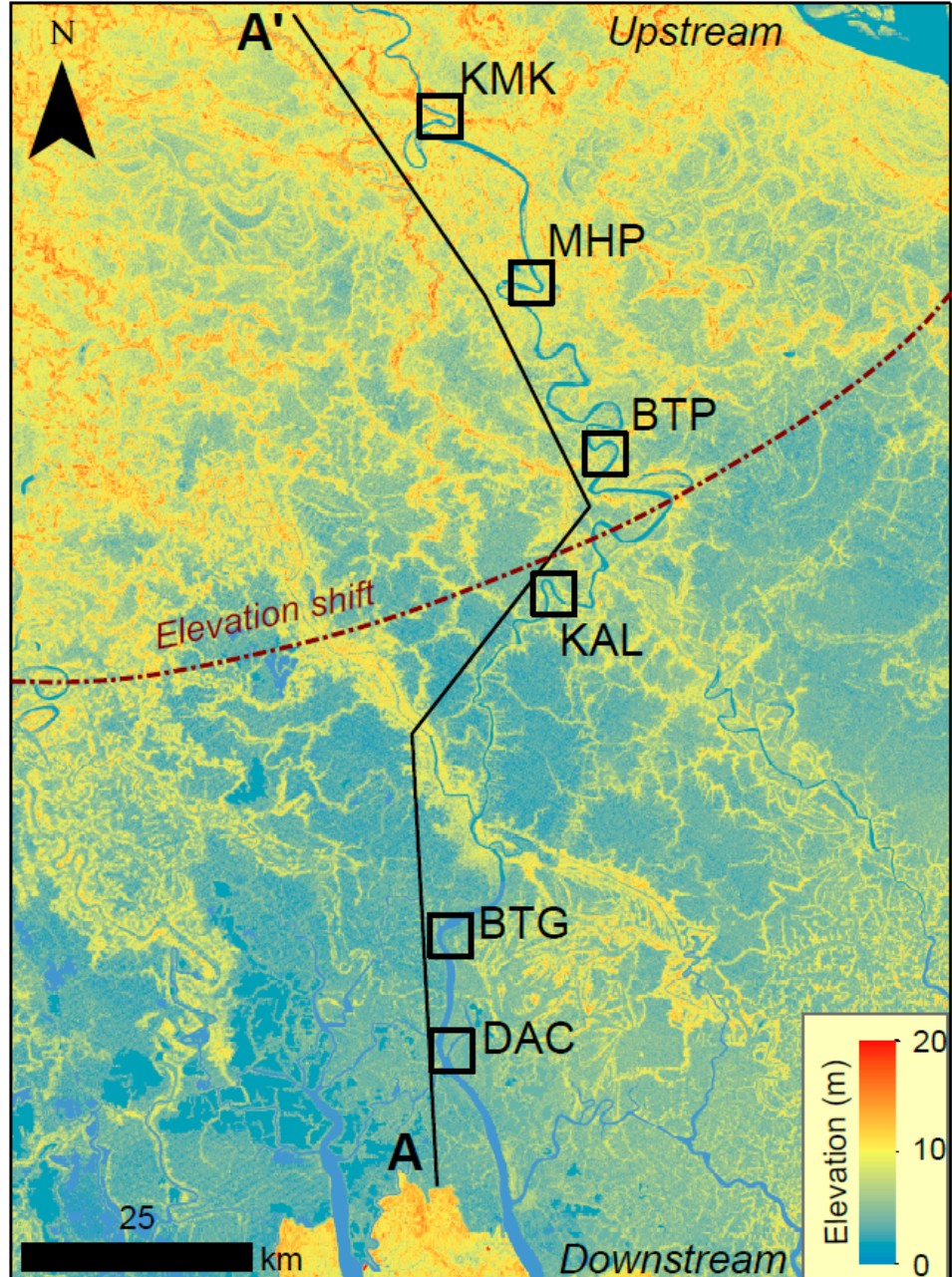

**Figure 2.** Digital elevation model (DEM) of southwest Bangladesh constructed from the 2000 Shuttle Radar Topography Mission data illustrating a distinct shift in topography. Transect A-A' runs alongside the Gorai River from the northern edge of the Sundarbans Forest to the Ganges-Gorai offtake.

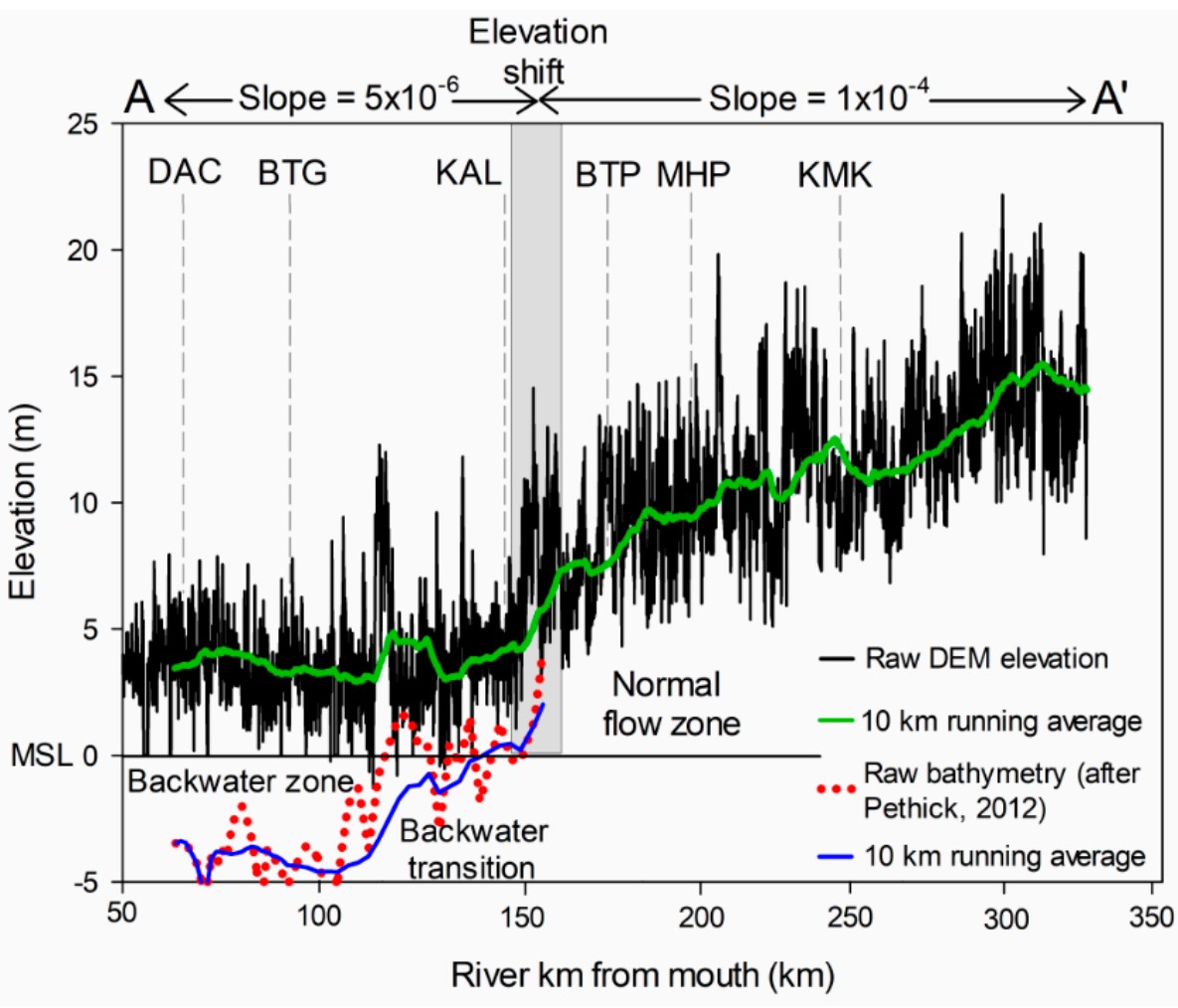

**Figure 3.** Floodplain elevation measured adjacent to the Gorai River, following transect A-A′ of Figure 2. The vertical position of the channel bed, displayed in blue relative to mean sea level (MSL), is calculated as the 10 km running average of the floodplain elevation minus the 10 km running average of the channel depth (after Pethick [56]). Note that the transition from backwater to normal flow hydrodynamics occurs immediately downstream of KAL and the elevation shift.

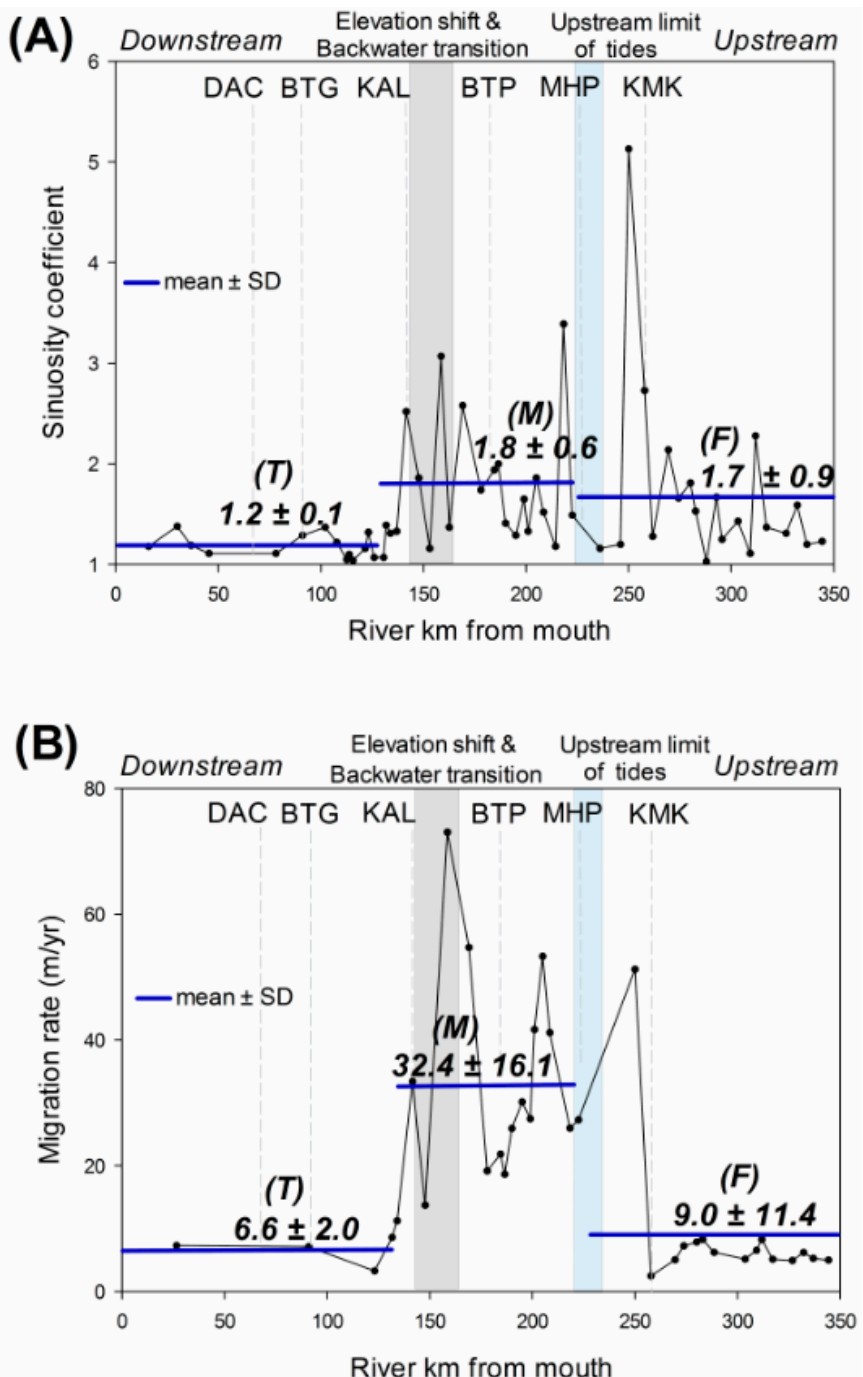

**Figure 4.** Trends in channel morphodynamic properties as a function of distance from the river mouth, including: (**A**) present-day channel sinuosity and (**B**) meander bend migration rates between 1972 and 2015. Average ± standard deviation (SD) values for tidal (T), mixed tidal-fluvial (M), and fluvial (F) hydrodynamic zones are indicated by blue horizontal lines. The elevation shift and backwater transition are displayed as a shaded envelope because they occur over an area of the river rather than at a single point. Similarly, the upstream limit of tidal influence is represented as an envelope because the limit depends on the magnitude of river discharge.

## 4.2. Shallow Subsurface Sedimentology and Stratigraphy

### 4.2.1. Tide-Dominated Depozone (DAC & BTG)

Dacope (DAC), the most downstream core site, is located 70 river km upstream of the mouth and 60 river km downstream of the backwater limit (Figures 1B and 3). The mean tide range of this area is estimated to be ~3.2 m based on tide gauge data at Mongla (~3.1 m, [58]), located 15 river km downstream of DAC. The stratigraphy of DAC cores is dominantly composed of medium silt (72.3%) with occasional coarse silt (15.7%) and fine silt beds (12.0%, Table 1; Figure 5). Down-core grain size trends display either weak reverse grading (i.e., coarsening upward, see DAC-D, Figure 5) or are invariant with depth (e.g., DAC-U, Figure 5). Although core profiles appear homogeneous, x-radiographs reveal that the sedimentary fabric is composed of rhythmic, mm-scale laminations of medium silts (Figure 6A). Apart from human influences, bioturbation is minimal and much of the original sedimentary fabric is present (bioturbation index, BI = 0–2, Figure 6A).

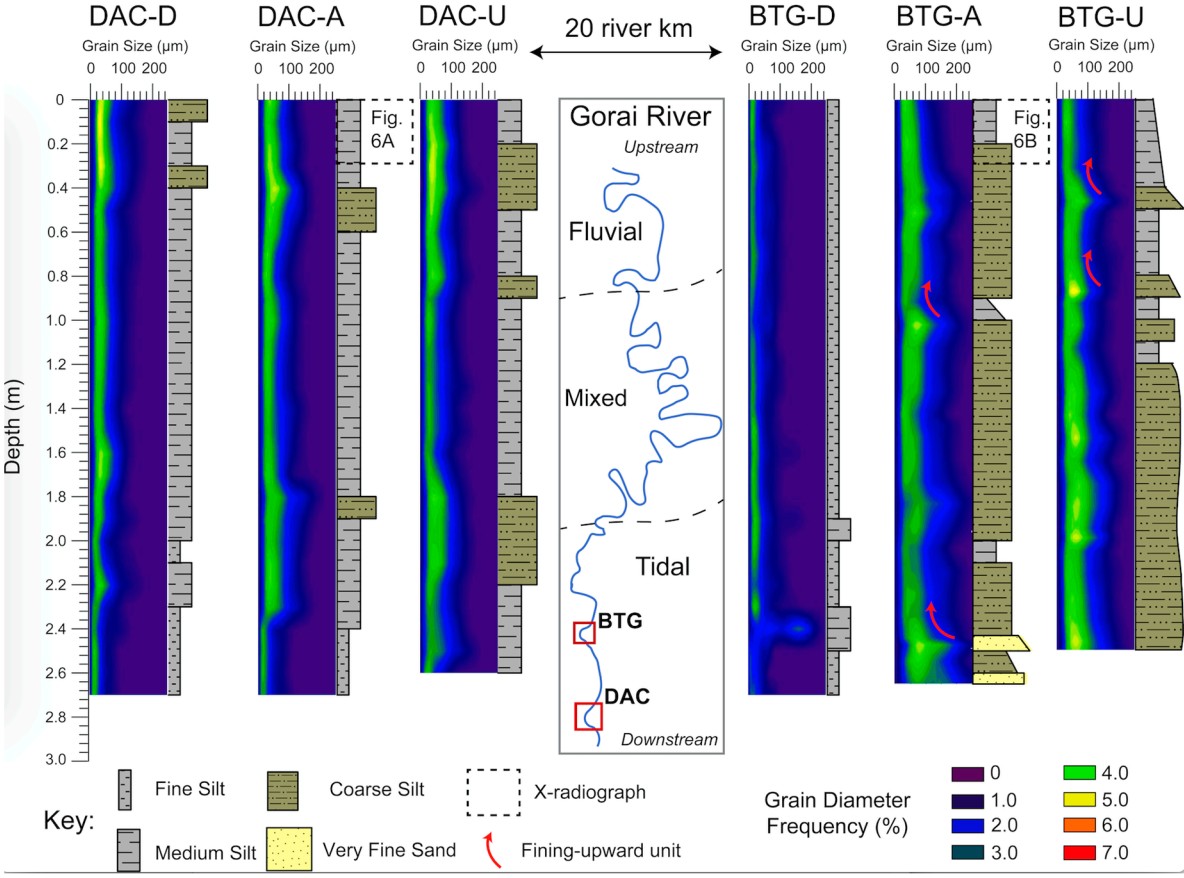

**Figure 5.** Core stratigraphy at Dacope (DAC) and Batiaghata (BTG) sites, located within the tidal depositional zone (mean tide range at DAC and BTG are 2.8 m and 2.4 m, respectively). Down-core trends in grain size are represented by two panels for each core: volumetric frequency distributions between 0 and 250 μm (**left**), and median grain size (**right**).

**Table 1.** Median grain size proportions for auger core locations and depositional zones. Note that changes in sand (and mud) content are particularly pronounced between BTG and KAL (across the backwater), as well as between MHP and KMK (near the limit of tidal influence). Mean tide ranges, originally reported by EGIS (2000) and BIWTA (2019), take into account neap/spring variation and seasonal conditions. Water salinity values for the dry season were reported by Winterwerp and Giardino (2012); water salinity is negligible for all sites during the monsoon season.

| TFT Hydrodynamic Conditions | Tide-dominated | Tide-dominated | Mixed Tidal-fluvial | Mixed Tidal-fluvial | Fluvial-dominated | Fluvial-dominated |
|---|---|---|---|---|---|---|
| Auger core location/ Grain size (%) | DAC | BTG | KAL | BTP | MHP | KMK |
| Fine-medium silt (8–32 μm) | 84.3 | 36.5 | 31.5 | 11.8 | 7.0 | 0 |
| Coarse silt (32–63 μm) | 15.7 | 51.2 | 41.6 | 54.1 | 37.2 | 13.3 |
| Very fine sand (63–125 μm) | 0 | 2.3 | 26.9 | 32.9 | 45.3 | 84.5 |
| Fine sand (125–250 μm) | 0 | 0 | 0 | 1.2 | 10.5 | 2.2 |
| Sand : mud ratio | 0 : 100 | 2.3 : 97.7 | 26.9 : 73.1 | 34.1 : 65.9 | 55.8 : 44.2 | 86.7 : 13.3 |
| Mean tide range (m) | 3.2 | 2.9 | 1.8 | 1.4 | 0.2 | 0 |
| Water salinity, dry season (psu) | 23.0 | 22.0 | 10.0 | 4.0 | 0.5 | 0 |

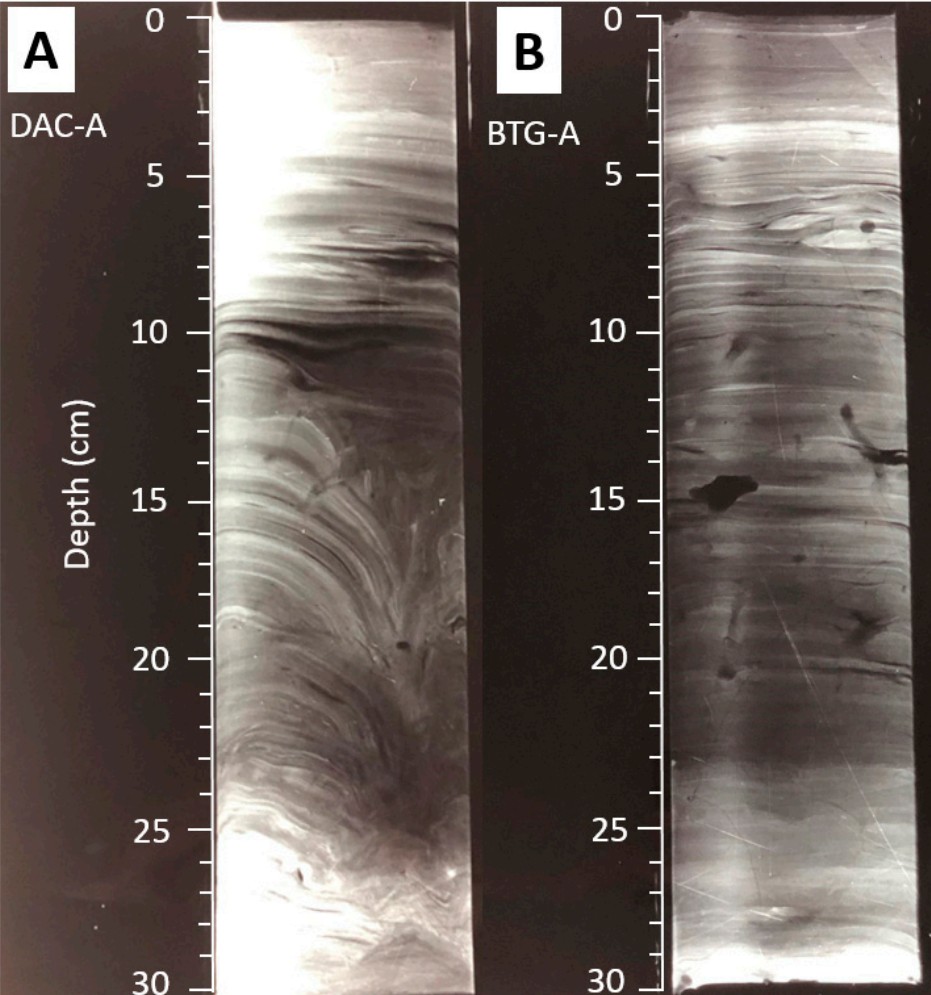

**Figure 6.** X-radiograph negatives collected from box cores at (**A**) DAC-A and (**B**) BTG-A locations illustrating evidence of tidal and seasonal depositional influences (see Figure 5 for core locations and position within the stratigraphy). Darker colors represent less dense sediments (e.g., fine silts), while light colors correspond to sediments with relatively high density (e.g., coarse silts and sands). Extensive deformation of bedding below 10 cm depth at DAC-A was likely produced by human activity (footprint).

Batiaghata (BTG) is situated 20 river km upstream of DAC and 40 river km downstream of the backwater limit (Figures 1B and 3). The local tide range is estimated to be ~2.9 m based on tide gauge data at Khulna (~2.3 m [62]) and Mongla (~3.1 m [58]). The stratigraphy of apex and upstream cores on this point bar consists of moderately well-sorted coarse silt with occasional medium silt and rare very fine sand beds (Figure 5). Sediments of the downstream core are comparatively finer, dominantly composed of well-sorted fine silt (Figure 5). All BTG cores exhibit gradual normal grading but rarely depart from the median grain size (Figure 5). Due to the relative invariance in grain size, the thickness of depositional packages is usually unclear. However, in a few cases, fining-upward units of ~20 to 30 cm thickness are observed (e.g., BTG-U, 0.3-0.5 m depth; Figure 5). X-radiographs indicate that the sedimentary fabric is characterized by sections containing rhythmic mm-scale laminations of silts (e.g., 5–10 cm depth and 12–17 cm depth, Figure 6B) as well as sections containing non-rhythmic, cm-scale laminations and beds (e.g., 3.5–4.5 cm depth and 10–12 cm depth, Figure 6B). Faunal burrows are occasionally observed (e.g., 18–19 cm depth, Figure 6B), but for the most part, the original sedimentary fabric is intact (BI = 0–2, Figure 6B). Overall, mud (grain size = 4–63 µm) composes 98.8% of sediments in the tide-dominated depozone (DAC and BTG, Table 1).

### 4.2.2. Mixed Tidal-Fluvial Depozone (KAL & BTP)

Kalia (KAL) is situated 50 river km upstream of BTG and 10 river km upstream of the backwater limit (Figures 1B and 3). The mean tide range of this area is inferred to be ~1.8 m based on tide gauge data at Khulna (~2.3 m [62]) and Lohagara (~1.5 m [63]), located 40 river km upstream of KAL. Sediment cores taken at KAL are chiefly composed of moderately well-sorted coarse silt (41.6%), with roughly equal proportions of medium silt (27%) and very fine sand (26.9%, Table 1; Figure 7). KAL cores contain a notably higher sand content (26.9%) in comparison to those at BTG (2.3%, Table 1). Individual beds are also more readily observable in KAL cores: KAL-D, for instance, contains at least eight fining-upward units that range from 0.2 to 0.5 m in thickness (Figure 7). Bed contacts are uniformly sharp at the base and fine upward either gradually (e.g., KAL-D, 1.0–1.3 m depth; Figure 7) or abruptly (e.g., KAL-A, 1.3–1.5 m depth; Figure 7). On the whole, fining-upward successions do not occur in consistently spaced intervals (Figure 7). In contrast to DAC and BTG, no sedimentological trends or structures indicative of tidal influence were seen in the stratigraphy of KAL cores.

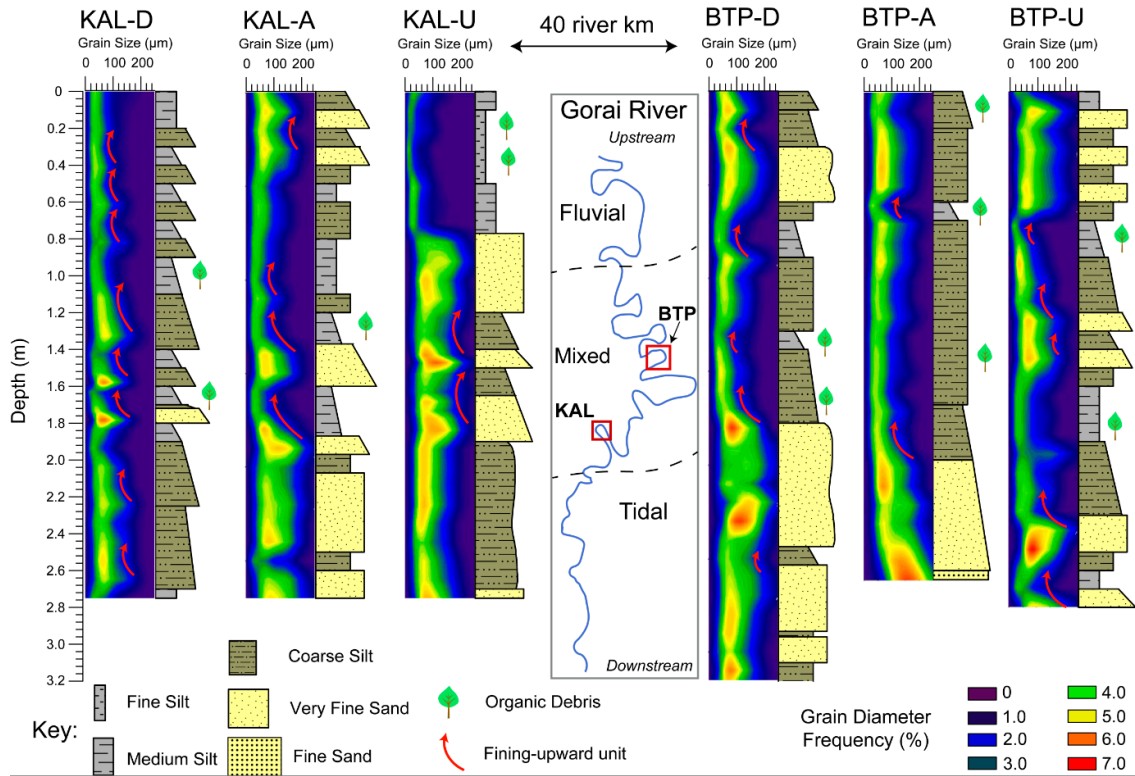

**Figure 7.** Core stratigraphy at Kalia (KAL) and Bhatiapara (BTP) locations, both of which are situated within the mixed tidal-fluvial depositional realm (mean tide range at KAL and BTP are ~1.8 m and ~1.4 m, respectively). Note the abundance of dm-scale fining-upward units, which are interpreted to represent monsoonal flood deposits.

Bhatiapara (BTP) is located 45 river km upstream of KAL and 35 river km upstream of the backwater (Figs 1B, 3). The mean tide range of this area is estimated to be ~1.4 m based on tide gauge data at Lohagara (~1.5 m [63]), located 10 river km downstream of BTP. Cores at BTP are dominantly composed of coarse silt (54.1%) and very fine sand (32.9%) with lesser amounts of medium silt (11.8%, Table 1; Figure 7). Normal grading is evident in downstream and apex cores from this point bar: From the surface to ~1.8 m depth, the stratigraphy largely consists of medium and coarse silt beds, whereas the lowermost sections are predominantly composed of very fine sand beds (Figure 7). Conversely, the upstream core along this point bar does not exhibit any clear grain size trends with depth (Figure 7). Irregularly-spaced fining-upward units of 0.2–0.5 cm thickness, similar to those in KAL cores, are

seen in all BTP cores (e.g., BTP-D, 1.6–1.8 m depth; Figure 7). Disseminated organic debris, including stems, twigs, and other fine particulate plant matter, are often found in silt beds associated with fining-upward units in KAL and BTP cores (Figure 7). When partitioned by hydrodynamic zone, the mixed tidal-fluvial setting contains substantially more sand than the tidal setting (29.3% vs. 1.2%, Table 1).

### 4.2.3. Fluvial-Dominated Depozone (MHP & KMK)

MHP is located 40 river km upstream of BTP and 35 river km downstream of KMK (Figure 1B). This area is inferred to be in the vicinity of the upstream limit of tidal influence based on hydrologic data from BIWTA (2019) [58] and anecdotal evidence from local villagers who observe minor tide-induced water level changes (tide range ~ 0.2 m) during the dry season, but no tides during the monsoon [64]. MHP cores mainly contain very fine sand (45.3%) and coarse silt (37.2%), with roughly equal parts medium silt (7%) and fine sand (10.5%, Table 1; Figure 8). All cores exhibit normal grading with a ~1.2-m-thick section of very fine and fine sands overlain by ~1.5 m of silt and very fine sand (Figure 8). Bed contacts, especially in the apex and upstream cores of this point bar, tend to be very abrupt at the base (e.g., MHP-A, 1.5–1.6 m depth; Figure 8).

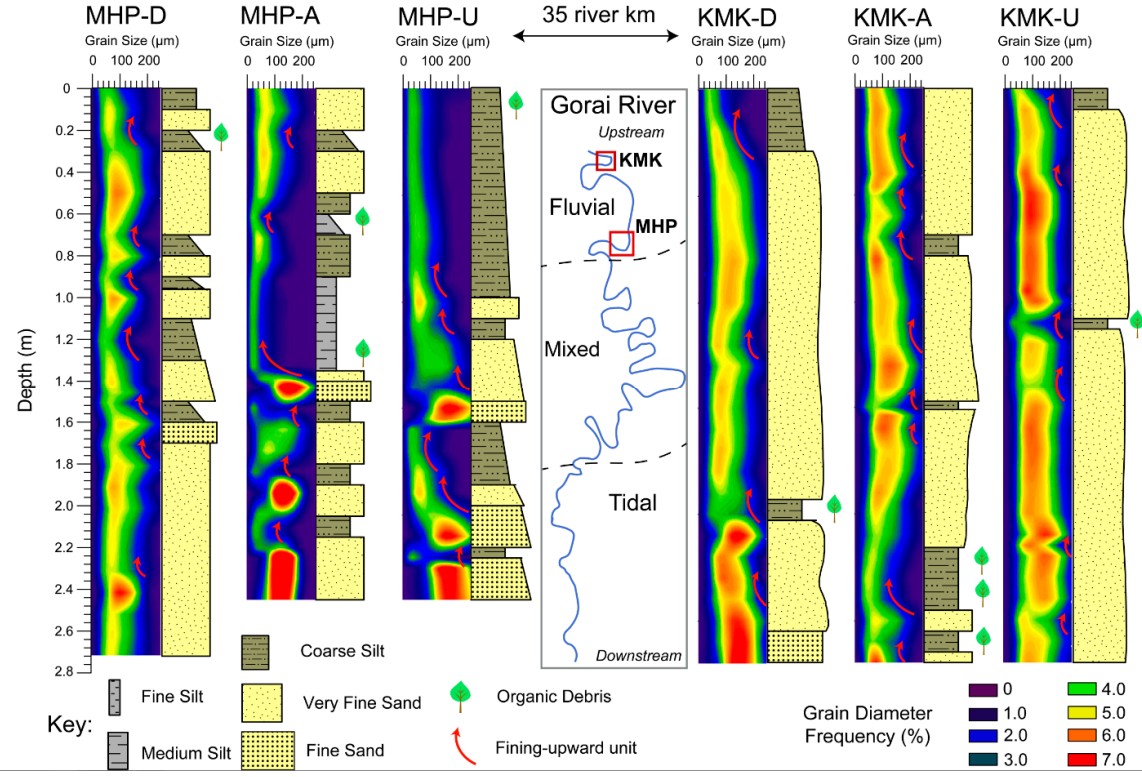

**Figure 8.** Core stratigraphy at Mohammadpur (MHP) and Kamarkhali (KMK), both of which are located within the fluvial depositional zone (mean tide range at MHP and KMK are ~0.2 m and 0 m, respectively). Note the thick (m-scale) and possibly amalgamated sand beds.

KMK, the most upstream core site, is situated 35 river km upstream of MHP and 30 river km upstream of the limit of tidal influence [58]. Cores from KMK are dominantly composed of very fine sand (84.5%), with lesser proportions of coarse silt (13.3%) and fine sand (2.2%, Table 1; Figure 8). The general stratigraphy of KMK cores consists of dm- to m-scale very fine sand beds interspersed with cm- to dm-scale coarse silt beds (Figure 8). Grain size fines upward, except for the core on the apex of the point bar (KMK-A, Figure 8). Silt beds are irregularly spaced throughout the core profiles and often contain decomposed organic litter (Figure 8). Sand beds appear to be massive

and structureless, though volumetric frequency distributions of grain size reveal fining-upward and occasional coarsening-upward successions (Figure 8). The difference in sand content between MHP (55.8%) and KMK (86.7%) represents the most pronounced change between two adjacent core sites in the study (Table 1).

### 4.3. Deep Subsurface Sedimentology and Stratigraphy

Drill cores from three point bar locations along the Gorai River (DAC, BTP, and KMK) extend stratigraphic profiles obtained by auger coring and demonstrate that along-channel stratigraphy varies substantially among the sites (Figure 9). A drill core from DAC indicates that the overbank muds observed throughout the auger cores in this location (Figure 5) continue to 13 m depth and overlie a 5-m-thick bed of well-sorted fine sands (Figure 9). Underlying this sand bed is a 17 m section characterized by alternating sands and muds (DAC-Drill, 17-34 m depth, Figure 9). Individual sand beds in this section are relatively thin (mean thickness ± standard deviation = 1.8 ± 0.4 m), fine upward, and contain particulate organic debris at the top of bed contacts (Figure 9). Mud beds are comparatively thicker (2.8 ± 2.0) and are composed of fine or medium silt (Figure 9). The deepest section recorded in the core is a 12 m layer of amalgamated sand beds (DAC-Drill, 34–46 m depth, Figure 9). Overall, the abundance of mud (47.8%) and sand (52.2%) is roughly equal throughout the core (Table 2), though muddy units are concentrated in the upper half of the core profile (Figure 9).

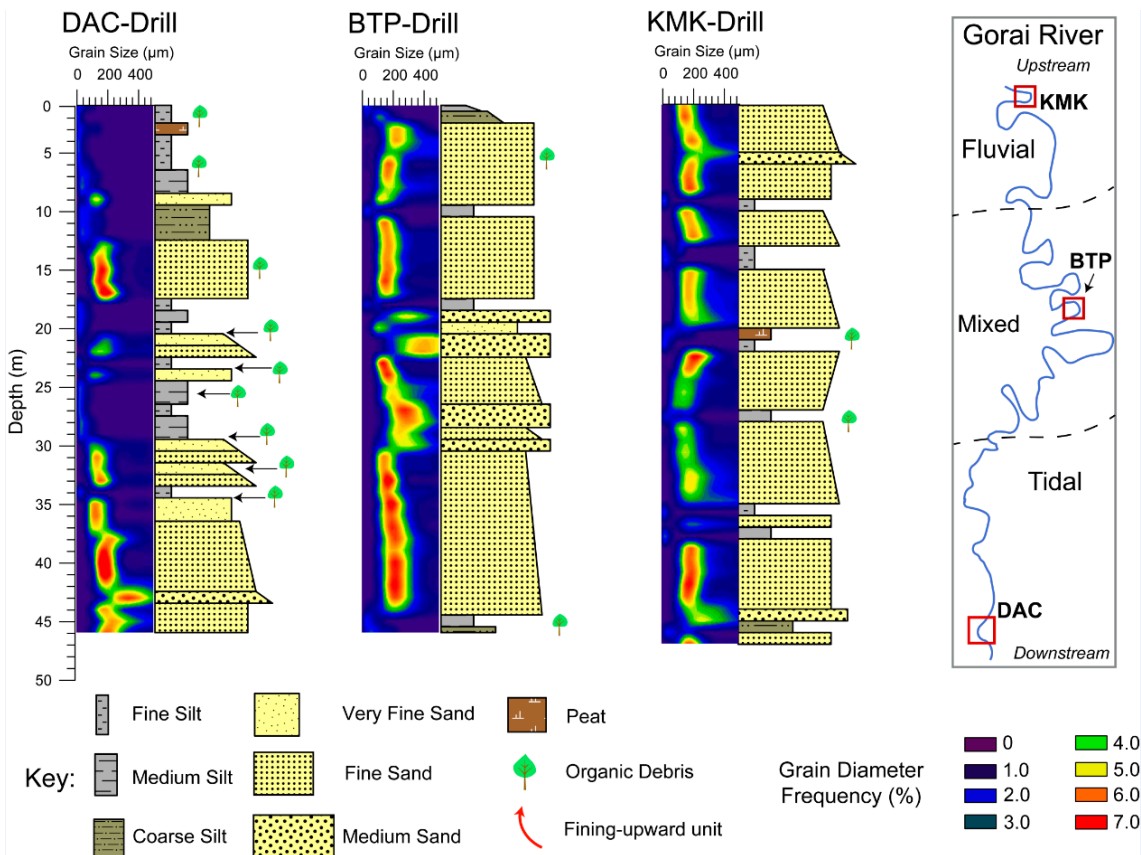

**Figure 9.** Drill core stratigraphy at DAC, BTP, and KMK point bars, representing tidal, mixed tidal-fluvial, and fluvial depositional settings. Note the change in scale for core depth relative to cores shown in Figures 5, 7 and 8.

**Table 2.** Grain size proportions for drill core locations.

| TFT Hydrodynamic Conditions | Tide-dominated | Mixed Tidal-fluvial | Fluvial-dominated |
|---|---|---|---|
| Drill core location/Grain size (%) | DAC | BTP | KMK |
| Peat | 2.2 | 0 | 2.1 |
| Fine-medium silt (8–32 μm) | 35.6 | 6.7 | 14.9 |
| Coarse silt (32–63 μm) | 6.7 | 4.4 | 2.1 |
| Very fine sand (63–125 μm) | 15.6 | 2.2 | 0 |
| Fine sand (125–250 μm) | 37.7 | 71.1 | 76.6 |
| Medium sand (250–500 μm) | 2.2 | 15.6 | 4.3 |
| Sand : mud ratio | 55.5 : 45.5 | 88.9 : 11.1 | 80.9 : 19.1 |

The stratigraphy at BTP contains considerably more sand in comparison to that of DAC (89.2% vs. 52.2%, respectively, Table 2), as suggested by auger core results (Figures 5 and 7). Sand beds at BTP are relatively thick (5.7 ± 4.0 m), well sorted, and often display fining upward trends (e.g., BTP-Drill, 22–29 m depth, Figure 9). Fine sand is the most common grain size of these beds, but lesser amounts of very fine and medium sand are also present (Table 2; Figure 9). Mud beds are thin (1.5 ± 0.5 m) and relatively scarce throughout the stratigraphy (Figure 9).

The stratigraphy of KMK, like BTP, is dominantly composed of well-sorted fine and medium sands (combined = 80.9%, Table 2; Figure 9). The primary difference between these two sites is that the stratigraphy of KMK exhibits a greater abundance of interspersed mud beds (7 at KMK vs. 4 at BTP, Figure 9), a quality that is also reflected in the overall proportion of mud in the stratigraphic profiles (19.1% at KMK vs. 10.8% at BTP, Table 2).

## 5. Discussion

### 5.1. Sedimentological Trends across the TFT

Core data from point bars across the Gorai River TFT reveal three sedimentological and stratigraphic trends that reflect the relative influence of fluvial and tidal hydrodynamics and associated depositional processes. The three recognized trends correspond to: (1) down-core changes in grain size, (2) the extent and type of bedding exhibited, and (3) the relative abundance of sand and mud preserved in the stratigraphy.

5.1.1. Down-core Trends in Grain Size

Sediment cores collected from tide-dominated point bars (DAC and BTG) do not exhibit any consistent vertical changes in grain size, demonstrating fining-upward (BTG-A and BTG-U), coarsening-upward (DAC-D and DAC-A), and invariant trends (DAC-U and BTG-D) (Figure 5). These observations are in line with conflicting reports in the literature of fining-upward [18,65,66] and coarsening-upward [29,67] successions occurring in tidal point bars. Detailed research on a tidal point bar in Venice Lagoon, Italy, indicates that asymmetries in ebb- and flood-tide flow velocities can construct complex stratigraphic architecture, including the presence of fining- and coarsening-upward sequences in the same point bar [68]. We postulate that similar hydrodynamic conditions lead to the inconsistent down-core grain size trends observed in the present study (Figure 5), especially in light of recent work demonstrating that the Ganges-Brahmaputra tidal delta plain is characterized by a tidal prism that is both seasonally and directionally asymmetric [69]. Localized geomorphological elements, such as channel confluences, can also impact hydraulic conditions and resultant channel margin sedimentology (e.g., Refs. [57,70]). Both of the tidal point bars investigated in this study (DAC

and BTG) are situated adjacent to major tidal channel confluences (Figure 1B), which may generate irregular hydrodynamic conditions (e.g., eddying and rapid flow reversals [71]) and stratal architecture.

In the mixed tidal-fluvial realm, normal gradation is the primary vertical grain-size trend, occurring in four out of the six cores taken in this setting (KAL-A, KAL-U, BTP-D, and BTP-A; Figure 7). Cores that did not display normal gradation (KAL-D and BTP-U, Figure 7), were generally invariant with depth and may have exhibited normal grading if the cores were deep enough to reach the channel thalweg sands (see BTP-Drill, Figure 9). Normal gradation is a common element of fluvial point bar depositional models and is attributed to the progressive accumulation of finer overbank sediments over coarser thalweg sands as the channel migrates across the floodplain (e.g., Refs. [18,53,72,73]). Inherent to this model of formation is the notion that fluvial point bars are laterally mobile features, migrating over time through extension and down-valley translation (e.g., Refs. [54,74]). Thus, the prevalence of normal grading in the KAL and BTP cores (Figure 7), along with elevated rates of channel sinuosity and mobility (Figure 4), suggests that fluvial processes are, at least in part, operating in this hydrologic domain. Likewise, cores from the fluvial realm almost exclusively demonstrate normal gradation, with the only exception being KMK-A, which shows reverse gradation (Figure 8). The drill core from KMK contains multiple instances of m-scale fining-upward successions that may represent buried point bars (e.g., KMK-Drill, 8–13 m depth and 27–35 m depth, Figure 9), suggesting that similar hydrodynamic conditions have persisted in this region for extended periods of time (sensu Ref. [75]).

5.1.2. Progressive Changes in Bedding Type

The stratigraphy of point bars in the tidal realm (DAC and BTG) appears to be composed of texturally homogeneous muds with no readily distinguishable bedding or sedimentary structures (Figure 5; Supplementary Figure S1). However, x-radiographs reveal the presence of rhythmic, mm-scale laminations of mud ("tidal rhythmites," Figure 6), which reflect regular variations in hydraulic energy during the tidal cycle (e.g., Ref. [76]). Tidal rhythmites are commonly observed in tide-influenced sedimentary features, including point bars [18,19,65,77], in-channel bars [29], and channel fill deposits [20,25]. The non-rhythmic, cm-scale beds observed in the x-radiograph of BTG (Figure 6B) may have been deposited during time periods when tides are subordinate to river discharge, such as during the summer monsoon season. The preservation of original sedimentary fabric and low bioturbation index (BI = 0–2) at DAC and BTG (Figure 6) suggest rapid sedimentation (e.g., Ref. [78]) and a high degree of physico-chemical stress (e.g., alternating fresh and saline water conditions, [28,29,77]). These interpretations are in accordance with observational studies in the region, documenting sediment accretion rates in excess of 1 cm/yr [79–81] and seasonal differences in salinity of ~20 psu [41].

Further upstream in the mixed tidal-fluvial realm, bedding becomes visually apparent and is characterized by cm- to dm-scale fining-upward packages that are irregularly spaced throughout the stratigraphy (Figure 7, Supplementary Figure S2). On the basis of their thickness, normal gradation, abrupt basal contacts, and non-rhythmic nature, we interpret that the deposition of these units is controlled by seasonal changes in fluvial discharge. Specifically, we propose that sandy beds are deposited during monsoon conditions, while mud-rich beds are deposited during low flow conditions (i.e., dry season). Centimeter-scale sand beds (e.g., KAL-D, 1.75-1.80 m depth; Figure 7) likely represent sedimentation from a single monsoon flood pulse, as supported by sediment tile accumulation measurements [79,81]. Sand beds on the order of decimeters (e.g., BTP-D, 1.80–2.40 m depth; Figure 7) may indicate the amalgamation of several monsoon flood deposits, a possible product of time periods when repeatedly strong flood pulses eroded fine-grained low-flow deposits (sensu [16]). Mud-rich beds are homogeneous in texture (Figures 7 and 8, Supplementary Figure S2) with the exception of intercalated organic detritus, a common constituent of slackwater flood deposits [82]. Despite the presence of tides in the mixed tidal-fluvial realm (range ~ 0.5–2.0 m), any evidence of tidal influence in the stratigraphy has been completely overprinted by seasonal signals. This contrasts with findings in the Fraser River, where the co-presence of rhythmic and non-rhythmic heterolithic bedding was documented in point bars and in-channel bars in a mixed tidal-fluvial setting [29,37]. Stratigraphic

differences between these locations may reflect the overall importance of seasonal (e.g., Gorai River) vs. tidal (e.g., Fraser River) hydrodynamic conditions throughout the respective systems, applying to each mixed tidal-fluvial zone.

The abundance and greater thickness (i.e., m-scale) of sand beds in the stratigraphy of the fluvial realm (Figures 8 and 9, Table 1) suggests that most suspended load sediments (e.g. silts and clays) bypass this reach of the river and are conveyed downstream to regions of lower hydraulic gradient (Figures 2 and 3). Although fining-upward packages are detected in volumetric frequency distributions of the fluvial cores (e.g., KMK-A, 1.0–1.35 m depth; Figure 8), they are less commonly observed than in the mixed tidal-fluvial realm (Figure 7), which may be due to the greater erosive potential of monsoonal flood flows in the upstream tract of the river. In other words, monsoon flood flows may winnow relatively fine sediments from the point bar surface, ultimately yielding clean, well-sorted sands and a homogeneous stratigraphic profile over time (Figure 8, see also Ref. [53]).

### 5.1.3. Relative Abundance of Sand and Mud

Examination of grain size partitioning across the Gorai River TFT reveals that downstream changes in the relative proportions of sand and mud do not occur in a uniform manner, but rather exhibit two reaches with pronounced differences—the first occurring between BTG and KAL, where sand content decreases from 26.9% to 2.3%, and the second occurring between MHP and KMK, where sand content decreases from 86.7% to 55.8% (Figure 10; Table 1). We attribute the former shift in grain size to mass extraction of bedload sediments (i.e., sand) at the upstream limit of the backwater zone, which is situated immediately downstream of KAL (Figures 3 and 10). Backwater hydrodynamics have been shown to fundamentally influence sediment transport capacity in coastal rivers through the introduction of non-uniform water-flow conditions (e.g., Refs. [17,31,32,35,83,84]). For instance, Nittrouer (2013) [32] demonstrated in the Mississippi River, USA, that the rate of downstream fining of channel-bed sediments dramatically increases downstream of the backwater transition zone. We document a similar phenomenon in the present study wherein point bar sediments become notably coarser and sand-rich upstream of the backwater zone (Figures 5 and 7–9; Tables 1 and 2). The spatial coincidence of the backwater transition and coarsening of channel-bank particle size holds important implications for modern and ancient fluvial-tidal rivers. First, the preferential sequestration of bedload sediments induced by the backwater suggests that this reach of the river is a site of facilitated deposition and thus may be susceptible to channel-bed aggradation over time. This notion is supported by bathymetric data indicating marked shallowing of the channel bed between BTG and KAL (Figure 3). Second, many workers have hypothesized that channel-bed aggradation at the backwater dictates the location of avulsion nodes in fluvio-deltaic systems [43–46,85]. Although the Gorai River is not likely to avulse in the near future, we suggest that this may have occurred when the active lobe of the Ganges-Brahmaputra Delta occupied this region ~2000–4000 years before present [86]. Third, spatially constraining the paleo-backwater zone in ancient fluvial and tidal depositional environments is useful for predicting the quality of channel fill and point bar hydrocarbon reservoirs (e.g., sand-mud ratio, lateral continuity of beds, porosity and permeability). In a recent example, Martin et al. [17] generated amplitude extraction maps of interpreted channel belt horizons from a 3-D seismic dataset to identify the ancient planform geomorphology of the Triassic Mungaroo paleo-delta system. If a similar approach is taken, and the sediment source direction (i.e., provenance) can be determined (e.g., Ref. [87]), then the results of this study suggest that point bar deposits upstream of the paleo-backwater should be sought out to maximize the potential of encountering high-quality reservoir sands.

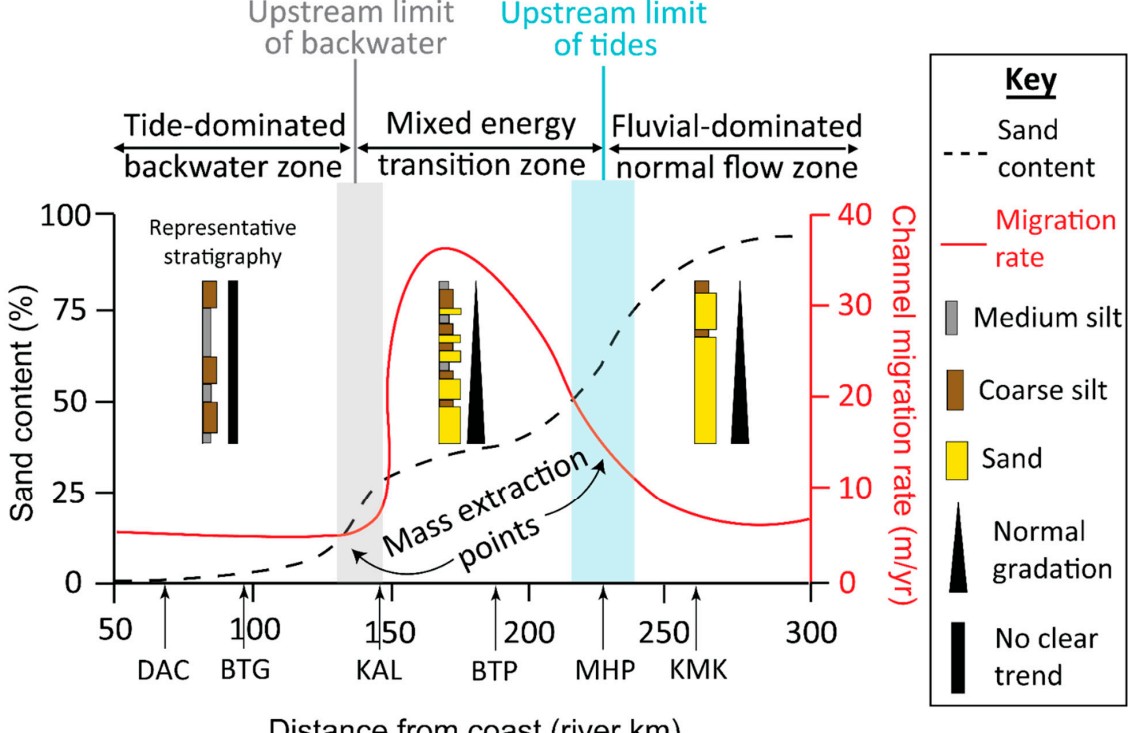

**Figure 10.** Integrated depositional framework for the tidal to fluvial transition of the Gorai River with representative core profiles for each depositional zone. The upstream extents of the backwater and tidal influence are not singular points in space or time (e.g., Dalrymple and Choi [1]), and as such, are represented as shaded envelopes. The gray shaded zone represents the backwater transition, where parts, but not all, of the channel bed are situated above mean sea level (see Figure 3). The blue shaded zone represents the limit of tidal influence, which varies depending on seasonal conditions, in particular, the magnitude of fluvial discharge.

The second shift in grain size, occurring between MHP and KMK (Figure 10, Table 1), is hypothesized to result from the landward extent of tidal influence. Similar to the backwater effect, non-uniform water flow conditions induced by tides (i.e., current reversals) are space- and time-dependent (e.g., Ref. [88]). Specifically, these hydraulic conditions only occur during landward-directed flood tides and are amplified during periods of low river discharge (e.g., Ref. [89]). Although the most landward extent of tide-induced current reversals does not necessarily overlap with the limit of tidal influence [21,27], the exceedingly low discharge of the Gorai River during the dry season (~10 m$^3$/s, compared to ~2000 m$^3$/s during the monsoon, [41]) indicates that flood tides experience little resistance from fluvial forces and therefore likely induce changes in current direction near the tidal limit. Previous studies on sediment grain size partitioning in rivers have examined microtidal systems (e.g., Mississippi River [35]; Trinity River [84]) to focus on the effects of the backwater and avert instances of non-uniform flow induced by tides. In these cases, one distinct shift in grain size was noted, which was coincident with the upstream limit of the backwater (e.g., Ref. [32]). Based on the results of this study, we suggest that tide-influenced rivers may display two separate shifts in sediment grain size—the first occurring near the backwater and the second occurring near the landward extent of current reversals induced by tides (Figure 10), with the expectation that the distance between these two sedimentological shifts is highly site-specific.

*5.2. Channel Morphodynamics across the TFT*

The planform expression and recent (i.e., decadal) evolution of the Gorai River indicate a downstream-to-upstream pattern of channel morphodynamics characterized by: (i) non-sinuous

and relatively immobile in the tidal realm, (ii) sinuous and mobile in the mixed tidal-fluvial realm, and (iii) sinuous and relatively immobile in the fluvial realm (Figures 1, 4 and 10). A similar "straight-meandering-straight" channel pattern was initially described by Dalrymple et al. [90] and is often associated with tide-dominated estuaries [1]. Although the Gorai River is not an estuary in a strict sense, it functions much like one during the dry season when water discharge is reduced to ~10 m$^3$/s [41] and flood tides take precedence over ebb tides [69].

In the downstream tide-dominated depozone, uniformly low sinuosity and channel migration rates (Figures 4 and 10) likely result from a combination of factors inherent to this realm of the TFT, including relatively low hydraulic gradient and flow strength (Figure 2; see also Ref. [53]) and an abundance of fine-grained, cohesive sediments (Figures 5, 9 and 10; Tables 1 and 2). Indeed, mud, which composes 98.8% of point bar sediments in the tidal realm (Table 1), has been shown to restrict lateral channel mobility both in physical experiments (e.g., Refs. [91,92]) and observational studies (e.g., Refs. [74,93–95]) owing to its relatively high critical shear stress (e.g., Ref. [96]). Low channel sinuosity and mobility may also be ascribed to the predominance of tidal hydrodynamics [97,98]. For instance, Lentsch et al. [98] demonstrated through a series of physical delta models that distributary channel mobility systematically decreases as the ratio of tidal to fluvial energy increases. Enhanced channel stability in tide-dominated settings was attributed to channel deepening and sediment flushing via ebb tide-enhanced river flow [98]. These findings are consistent with observations of the present study in that channel depth markedly increases (Figure 3), while sinuosity and migration rates decrease downstream of KAL (Figure 4), the reach of the Gorai River where tides are the predominant hydraulic force [50].

An abrupt increase in channel mobility and sinuosity occurs immediately upstream of the tide-dominated realm and persists throughout the mixed tidal-fluvial realm (Figures 4 and 10). This shift in river kinematics coincides with the upstream limit of the backwater zone (Figures 3 and 4), suggesting an interconnected relationship among changes in hydrodynamics, bank sedimentology, and channel morphodynamics (e.g., Ref. [34]). In particular, we suggest that this marked change in channel mobility arises from backwater-induced modification of sediment composition and sediment storage (sensu Ref. [35]). At the upstream limit of the backwater (Figure 3), the capture and preservation of non-cohesive sediments in channel margin bodies (Figure 6, Table 1) facilitates bank erodibility and therefore channel mobility across the floodplain (Figure 4; see also Ref. [94]). Augmented sediment storage and lateral building of point bars upstream of the backwater also drives fluvial erosion by constricting channel flow and redirecting high-energy flows of the channel thalweg toward the cutbank [99,100].

Stratigraphic profiles at KMK indicate an abundance of non-cohesive, sandy sediment (Figures 8–10), a feature that presumably applies to channel banks upstream of KMK. Yet, lateral channel migration rates conspicuously decrease upstream of KMK to values similar to those of the tidal realm (Figures 4B and 10). EGIS [50] speculated that this relatively straight and immobile reach of the Gorai is caused by the older and more consolidated substrate of the upper Ganges floodplain. An alternative explanation for restricted channel mobility in this area could be an abundance of subsurface clay plugs that formed following meander cutoff and oxbow lake infilling. Indeed, similar upstream decreases in channel migration rates have also been observed in the Mississippi River and attributed to the abundance of clay plugs [101,102]. Deep drill cores from this study indicate nearly twice as many intercalated mud units in the stratigraphy of KMK when compared to BTP (Figure 9). Similar findings have been noted in cores collected from the floodplain of the Gorai River [75]. For example, a drill core taken at Magura, which is situated ~8 km southwest of KMK (see Figure 1B for location), documented stratigraphy consisting of roughly equal proportions of channel sands and floodplain muds [75]. In contrast, a drill core taken at Narail, located ~20 km southwest of BTP (see Figure 1B for location), was almost exclusively composed of fine and medium sands [75,103]. Taken together, the similarities of the deep stratigraphic profiles immediately adjacent to the Gorai River (this study) and

in the floodplain [75] suggest that if substrate composition is a primary control on channel morphology, then the morphodynamics should operate in a similar fashion as the river continues to shift over time.

## 6. Conclusions

Careful integration of remote sensing, river bathymetry, and core sedimentology and stratigraphy reveal insights on the processes that govern the tidal-fluvial transition of the Gorai River, southwest Bangladesh. The conclusions of this study are summarized as follows:

(1) The Gorai River can be divided into three tracts with distinct hydrodynamic and morphological properties: (i) a downstream tract dominated by meso-scale tides (range ~ 2–4 m) that exhibits uniformly low channel sinuosity and migration rates; (ii) a mixed tidal-fluvial tract affected by micro-scale tides (range < 2 m) and seasonal flood pulses that display high channel sinuosity and migration rates; and (iii) an upstream tract dominated by fluvial processes with little to no tidal influence (range < 0.2 m) that demonstrates moderate channel sinuosity and low migration rates.

(2) Point bar sedimentary architecture differs substantially among the three depositional zones. Point bars from the tide-dominated depo-zone are characterized by cohesive muds with no readily discernable bedding and no consistent vertical (i.e., down-core) grain-size trends. However, x-radiographs reveal rhythmic, mm-scale laminations (tidal rhythmites) and occasional cm-scale beds. The stratigraphy of the mixed tidal-fluvial realm is primarily composed of non-rhythmic, cm- to dm-scale sand beds that are interbedded with dm-scale mud beds containing particulate organic matter. These sand beds, which display erosive basal contacts and fine upward, are interpreted to represent monsoonal flood deposits. In the fluvial depositional realm, point bars consist of dm- to m-scale sand beds interbedded with organic-rich mud and almost exclusively display normal gradation.

(3) Comparisons of river channel bathymetry with elevation control of the adjacent floodplain indicate that the channel bed approximates mean sea-level ~135 river-km inland of the coast. This location, also known as the upstream limit of the backwater zone, may be a site of accelerated in-channel deposition due to the introduction of non-uniform water-flow conditions and resultant sequestration of bedload sediments. The spatial coincidence of the backwater limit and shift in river channel morphology suggests that this change in hydrodynamics may also locally influence the kinematics of the channel over time.

(4) Two pronounced shifts in point bar bulk sedimentology are observed along the course of the tide-influenced Gorai River: the first occurs between KAL and BTP and is spatially coincident with the upstream limit of the backwater zone (i.e., persistent non-uniform water-flow conditions); the second occurs between MHP and KMK and broadly coincides with the upstream limit of tidal influence (i.e., transient non-uniform water-flow conditions).

**Supplementary Materials:** The following are available online at http://www.mdpi.com/2073-4441/11/10/2047/s1, Figure S1: Section of a sediment core taken from a tidal point bar (DAC-D, 100-140 cm depth). Figure S2: Section of a sediment core taken from a mixed tidal-fluvial point bar (KAL-A, 130–150 cm depth).

**Author Contributions:** Conceptualization, E.J.B.; Data curation, E.J.B.; Formal analysis, E.J.B.; Funding acquisition, E.J.B. and C.A.W.; Investigation, E.J.B.; Project administration, E.J.B.; Resources, C.A.W. and D.K.D.; Supervision, C.A.W. and D.K.D.; Validation, C.A.W. and D.K.D.; Visualization, E.J.B.; Writing—original draft, E.J.B.; Writing—review & editing, E.J.B., C.A.W. and D.K.D.

**Funding:** This work was supported by research grants from the American Association of Petroleum Geologists (AAPG), the Geological Society of America (GSA), and the Society for Sedimentary Geology (SEPM). Additional funding was provided through National Science Foundation (NSF) Coastal SEES grant #1600258.

**Acknowledgments:** The authors kindly thank Arifur Rahman, Sourov Bijoy Datta, and Zahid Shawon for assistance in the field and for translating dialogue between EJB and local villagers. Special thanks to John Pethick for sharing bathymetric data for the lower Gorai River and to the Khulna Health Diagnostic Center for generating and developing x-radiographs for our non-medical purposes.

**Conflicts of Interest:** The authors declare no conflicts of interest.

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
