# Peer review of "An Integrated Approach for Constraining Depositional Zones in a Tide-Influenced River: Insights from the Gorai River, Southwest Bangladesh"

_water, doi:10.3390/w11102047_

Round 1
Reviewer 1 Report
General comment:
The present paper is of high quality from both scientific point of view and presentation point of view. The sediment characteristics and transport processes in the tide-influenced Gorai River (Bangladesh) are described in a very detailed and precise way. The study is mainly focussed on the interaction between fluvial and marine processes.
Specific comments:
Line 235: First, at this point, it should be explained that BI means Bioturbation Index.
References: The full titles of the journals should be given (if it is allowed by the author guidelines).
See annotated manuscript!

Author Response
Response to Reviewer #1
Dear Reviewer #1, thank you for your thoughtful evaluation of our manuscript and the suggestions that you provided. Please see our responses (in bold) to your specific comments (unbolded) below:
Specific comments:
Line 235: First, at this point, it should be explained that BI means Bioturbation Index.
Thank you for this suggestion – we have indicated that “BI” means “bioturbation index” in the revised manuscript.
References: The full titles of the journals should be given (if it is allowed by the author guidelines).
Thank you for this suggestion – while we agree that including the full titles of journals in the references is preferable, Water requires a specific reference style that uses abbreviations for some journals. Please see the Water guidelines for authors (under “back matter”) for more information: https://www.mdpi.com/journal/water/instructions
See annotated manuscript!
We appreciate the helpful text edits and have addressed these in the revised manuscript.
Reviewer 2 Report
Hi all,
Interesting work. A lot of factual material.
Very well structured. I appreciate the ”red wire”
I would like to see the entire FIGURES....I was very hard to follow.....OUT of page...
Author Response
Response to Reviewer #2
Dear Reviewer #2, thank you for your thoughtful evaluation of our manuscript and the suggestions that you provided. Please see our responses (in bold) to your specific comments (unbolded) below:
Specific comments:
I would like to see the entire FIGURES....I was very hard to follow.....OUT of page...
Some of the figures, specifically those related to core stratigraphy (e.g. Figs. 5, 7, 8, 9), are oriented in landscape mode for better readability. It appears that the PDF created by the manuscript submission software forces all pages into portrait mode and cuts off any content that does not fit into portrait size. If the reviewer has the option to view the manuscript in DOC format, we would recommend doing so to see the full content of landscape-oriented figures. If not, we are happy to provide these figures in some other format (e.g., TIF, EPS), if the editor allows it.